# A Scoping Review of Preterm Births in Sub-Saharan Africa: Burden, Risk Factors and Outcomes

**DOI:** 10.3390/ijerph191710537

**Published:** 2022-08-24

**Authors:** Adam Mabrouk, Amina Abubakar, Ezra Kipngetich Too, Esther Chongwo, Ifedayo M. Adetifa

**Affiliations:** 1KEMRI-Wellcome Trust Research Programme, Centre for Geographic Medicine Research (Coast), Kilifi P.O. Box 230-80108, Kenya; 2Department of Public Health, Pwani University, Kilifi P.O. Box 195-80108, Kenya; 3Institute for Human Development, Aga Khan University, Nairobi P.O. Box 30270-00100, Kenya; 4Department of Psychiatry, University of Oxford, Oxford OX3 7FZ, UK; 5Department of Infectious Diseases Epidemiology, London School of Hygiene and Tropical Medicine, London WC1E 7HT, UK; 6Department of Paediatrics, College of Medicine, University of Lagos, Idi-Araba, Lagos 100254, Nigeria

**Keywords:** preterm births, Sub-Saharan Africa, scoping review, burden, risk factors, outcomes

## Abstract

Preterm births (PTB) are the leading cause of neonatal deaths, the majority of which occur in low- and middle-income countries, particularly those in Sub-Saharan Africa (SSA). Understanding the epidemiology of prematurity is an essential step towards tackling the challenge of PTB in the sub-continent. We performed a scoping review of the burden, predictors and outcomes of PTB in SSA. We searched PubMed, Embase, and three other databases for articles published from the database inception to 10 July 2021. Studies reporting the prevalence of PTB, the associated risk factors, and/or its outcomes were eligible for inclusion in this review. Our literature search identified 4441 publications, but only 181 met the inclusion criteria. Last menstrual period (LMP) was the most commonly used method of estimating gestational age. The prevalence of PTB in SSA ranged from 3.4% to 49.4%. Several risk factors of PTB were identified in this review. The most frequently reported risk factors (i.e., reported in ≥10 studies) were previous history of PTB, underutilization of antenatal care (<4 visits), premature rupture of membrane, maternal age (≤20 or ≥35 years), inter-pregnancy interval, malaria, HIV and hypertension in pregnancy. Premature babies had high rates of hospital admissions, were at risk of poor growth and development, and were also at a high risk of morbidity and mortality. There is a high burden of PTB in SSA. The true burden of PTB is underestimated due to the widespread use of LMP, an unreliable and often inaccurate method for estimating gestational age. The associated risk factors for PTB are mostly modifiable and require an all-inclusive intervention to reduce the burden and improve outcomes in SSA.

## 1. Introduction

Worldwide, approximately 15 million births occur too early i.e., before 37 completed weeks of gestation every year [1]. These preterm or premature births (PTB) represent about 11% of all deliveries. However, there is significant health inequity with the proportion of PTBs varying across the different regions of the world—ranging from 5% in high income European countries to 18% in the low-income countries of South Asia and Sub-Saharan Africa (SSA) which are thought to have the highest burden of PTB [1]. In fact, SSA is home to nine of the eleven countries estimated to have the highest PTB burdens [1].

Prematurity is the leading cause of neonatal mortality, and with the recent improvements in under-five-mortality, it is now also the leading cause of childhood mortality in the first five years of life [2]. The target of the third Sustainable Development Goal (SDG3) is to reduce neonatal mortality to 12 deaths per 1000 live births and under five mortality rate to 25 deaths per 1000 live births by 2030 [3]. These targets can be achieved with the reduction in PTB-related mortality. However, progress in reducing neonatal mortality attributable to prematurity has stagnated and available evidence suggests that premature deliveries have been on the rise for the last two decades [4].

Babies born prematurely have a higher risk of adverse outcomes in childhood and adult survivors face long-term health problems affecting the brain, the cardiopulmonary system, hearing and/or vision [5,6,7]. Advances in obstetric and neonatal care, even in Low- and Middle-Income Countries (LMICs), have decreased the associated mortality and lowered the limit of viability [5]. However, the chances of survival of preterm babies is associated with the region of their birth [7]. Understanding the regional variations in risk factors is important to improve outcomes which will need the use of approaches that are responsive to each region’s specific challenges and limited resources.

The cause of PTB is multi-factorial with poverty, demographic, and obstetric factors all contributing but no specific cause is found in many cases. [8,9,10,11,12,13]. The distribution of these risk factors and their contributions to PTB is context-specific. Therefore, gaining an understanding of these factors is essential to the planning and deployment of prevention strategies for reducing the burden of PTB [7]. This is particularly important in SSA where the highest burden of PTB exist and health resources are limited. However, the epidemiology of PTB is poorly described in Africa for many reasons not limited to the challenge of dating pregnancies, lack of access to or poor uptake of antenatal care (ANC), a significant proportion of home deliveries without access to skilled care, the largely unknown contribution of stillbirths, and poor application of standardized definitions of early neonatal mortality [7]. In addition, the disparities in the management of prematurity within and across countries can contribute to the variable clinical outcomes seen [14].

The overall aim of this scoping review was to gain a better understanding of the current burden of PTB, associated maternal and infant risk factors, and the outcomes of PTB in SSA. We specifically conducted a scoping review because its broad nature was a good fit for our overall aim and for the anticipated heterogeneity of studies [15].

## 2. Materials and Methods

### 2.1. Overview

We followed the steps in the methodological framework first proposed by Arksey and O’Malley for conducting a scoping review [16]. In addition, we adhered to the recommendations in the guidelines for reporting Scoping Reviews (Preferred Reporting Items for Systematic reviews and Meta-Analyses extension for Scoping Reviews, PRISMA-ScR) [17].

### 2.2. Search Strategy and Study Selection

We searched PubMed, Embase, Web of science, African Journals Online and African Index Medicus for English language articles on PTB in SSA from database inception to 10 July 2021. The search strategy included the keywords “Preterm births” and “Sub-Saharan Africa” combined by the Boolean operator “AND”. Respective synonyms for these key words were combined using the Boolean operator “OR”. Appendix A provides the search string used in the PubMed database, which was modified to meet the specifications of the other databases. Additionally, we searched the reference list of the retrieved articles to identify additional papers relevant to our scoping review.

Search results were imported to EndNote X8™ (Bld10063) and duplicates were removed ahead of screening and selection. The studies were included if (i) they were conducted in SSA (ii) published in peer reviewed journals and in English language, and (iii) reported PTB burden, risk factors, and/or outcome regardless of study design. We excluded study protocols, commentaries, and conference abstracts. Study titles and abstracts were first reviewed for eligibility based on the inclusion and exclusion criteria described above by A.M. Full-text articles of selected abstracts were retrieved after which they were assessed for eligibility for inclusion in the review as well and was verified by I.M.A. and A.A. Updates, final check and verification before this submission were carried out by E.K.T. and E.C.

### 2.3. Data Extraction

The data extraction table was developed in Microsoft Excel (MS Excel 2016, Microsoft Corporation, Redmond, WA, USA) by A.M. and was refined in consultation with A.A. and I.M.A. To answer the review questions, we extracted and tabulated study characteristics (First author’s name, year of publication, country, population of study, and study design), measures of gestational assessment (GA) used, the reported PTB prevalence, risk factors, and outcomes PTB.

### 2.4. Synthesis of the Results

We grouped and summarised the reported proportions of PTB by the population and settings in which the studies were conducted. We determined the most common measures of GA used in SSA by summarising all the reported methods of estimating GA. We also narratively summarized the reported prevalence, risk factors and outcomes of PTB. In this review, we only considered the risk factors that were significant on multivariable analysis.

## 3. Results

### 3.1. Results of Database Search

The details of the search and selection process are as shown in Figure 1. The initial search in the bibliographic databases produced 4441 articles while 20 others were obtained from additional sources. A total of 181 studies meeting the inclusion criteria were included in the review.

### 3.2. Study Characteristics

The characteristics of included studies are shown in Appendix A. The included studies came from 24 countries in SSA (see Figure 2) with more than half of these (56.9%, *n* = 103) coming from only four countries (Ethiopia, *n* = 34; Nigeria, *n* = 25; South Africa, *n* = 24; and Tanzania, *n* = 20). Only a few (3.3%, *n* = 6) of these articles were multi-country studies. The included studies were published in the period from 1992 to 2021 with the majority (78.5%, *n* = 142) being published over the last decade (2011–2021) while about a half (50.8%, *n* = 92) of them were published in the last five years (2017–2021).

Most of the included studies were cohort studies. The rest were cross-sectional (45.9%, *n* = 83), and case-control studies (32.0%, *n* = 58). Only a few studies (5.5%, *n* = 10) were randomised controlled trials. Most of the included studies (69.6%, *n* = 126) recruited pregnant women from the general population. Forty studies (22.1%) recruited pregnant women from specific sub-populations such as adolescents and those with chronic illnesses such as HIV. In the remaining 15 of the included studies (8.3%), the target population were neonates admitted due to various conditions, including preterm neonates.

### 3.3. Methods of Gestational Age Assessment

The last menstrual period (LMP) method was the most used method of GA assessment among those studies that reported their methods of estimating GA. This method was exclusively used in 39 (21.5%) of the included studies. Ultrasound, which is regarded as the most accurate measure of GA, was exclusively used in 12 of the included studies (6.6%). Other methods of GA assessment that were used exclusively were the newborn maturational assessments methods, including, the Ballard score (*n* = 7 studies, 3.9%), the Dubowitz score (*n* = 3 studies, 1.7%), and the Finnstrom score (*n* = 1 study, 0.6%). Three of the included studies (9.9%) exclusively used the palpation-based method of estimating symphysis-fundal height. In 65 of the included studies, a combination of these methods was used. In these studies, LMP was the method used frequently in combination with other measures, with this method being used concurrently with other measures in 57 (87.7%) of these studies. In the remaining 51 studies (28.1%), the methods that were used to estimate GA were not reported.

### 3.4. Prevalence of PTB in Sub-Saharan Africa

In total, 133 studies reported the prevalence of PTB (Appendix A). Across studies, the reported prevalence estimates ranged from 3.4% to 49.4%. A hundred and eight studies assessed the prevalence of PTB among the general population of pregnant women and reported prevalence estimates ranging from 3.8% to 36.4%. Twenty-two studies assessed the prevalence of PTB in specific subpopulations of pregnant women and reported prevalence estimates ranging from 3.4% to 49.4%. Three of the studies that recruited sick neonates admitted for treatment reported the prevalence of PTB. The reported prevalence estimates of PTB in these studies were 16.5%. 21.4%, and 26.5%, respectively.

### 3.5. Risk Factors for PTB

Table 1 presents in detail the identified risk factors of PTB as reported in the included studies. In total, 97 of the included studies reported the risk factors of PTB in SSA. Across these studies, several factors were reported as predisposing women to preterm deliveries in different parts of SSA. Overall, these factors can be categorized into socio-demographic, obstetric, infections and/or co-morbidity, lifestyle, anthropometric, and treatment-related factors. Across all the categories, the most commonly reported risk factors for PTB (reported in ten or more of the included studies) were lack of or underutilization of antenatal care (ANC), maternal age, history of previous PTB, inter-pregnancy interval, malaria in pregnancy, hypertension in pregnancy, HIV in pregnancy, and premature rupture of membrane (PROM).

#### 3.5.1. Socio-Demographic Factors

In total, 38 studies reported the sociodemographic factors associated with PTB (Table 1). Maternal age was most commonly identified as a risk factor for PTB reported in 17 of these studies. Eight of these studies [11,21,62,64,66,80,98,106] reported an increased risk of PTB among younger mothers compared to older mothers although the age cut-offs for the younger mothers varied (≤16, ≤20 or ≤24). However, in one study [30], younger mothers aged ≤20 years were less likely to have a PTB. Seven studies [9,23,28,37,43,75,79] reported a significant association between advanced maternal age (>35 years) and risk of PTB. Six of these studies [9,23,28,37,43,75] reported increasing odds of PTB with advance in age. In contrast, Moodley et al. in South Africa [79], found a reduced risk of PTB among older women aged ≥35 years. In one study [93], pregnant women aged 20 to 34 years were reported to be at a higher risk of PTB.

Other reported sociodemographic factors included low socio-economic class [86,108], limited maternal education (only primary or no education) [77,95,104,106], rural residence [27,30,48,50,81,106], employment [41,86], Muslim mothers [41], mothers who had fear of delivery [41], and mother’s exposure to unfavourable working conditions including stressful jobs [41], exposure to vibration [12], and carrying heavy loads [24] which were all reported to increase the risk of PTB. Additionally, in 6 of the included studies [30,31,34,39,58,86], unmarried mothers had increased odds of having PTB compared to their married counterparts.

#### 3.5.2. Obstetric Factors

Forty-seven of the included studies reported the obstetric risk factors of PTB in SSA (Table 1). Lack of or underutilization of antenatal care (ANC) was by far the most reported obstetric risk factor associated with PTB. Twenty-six of the included studies [9,18,19,22,23,25,29,30,36,37,39,41,48,50,52,58,60,62,71,74,76,86,90,97,104,106] reported that pregnant mothers with fewer (<4 ANC visits) or with no ANC visits were more likely to have a PTB. In one study that compared the prevalence of PTB among mothers who had one, two, three or four ANC visits, the prevalence decreased with an increase in number of ANC visits (21.6%, 22.3%, 17.5% and 5.6%, respectively) [25]. In three of the included studies [9,48,52], an elevated risk of PTB was associated with lack of ANC services even when the baby was born in a health facility.

Previous history of PTB was found in 14 studies [9,19,26,27,54,58,61,68,90,91,94,101,104,110] to be significantly associated with increased risk of PTB in subsequent pregnancies. One study in Tanzania [68] evaluating the impact of previous PTB on successive deliveries found an increasing risk of PTB with an increase in the number of previous PTBs. In this study, compared with mothers who had term deliveries in their previous pregnancies, PTB risk increased among mothers who had their first PTB in the previous pregnancy (ARR = 2.7; CI = 2.1–3.4) and even further among mothers who had their second PTB in the previous pregnancy (ARR = 9.2; CI = 5.2–16.1) [68].

Premature rupture of membrane (PROM) was another frequently reported risk factor of PTB reported in 15 of the included studies [9,18,19,20,23,27,29,37,50,74,76,77,86,91,103]. In addition, other adverse birth outcomes in previous deliveries such as previous abortions [49,63,77,81,104], caesarean delivery [26,86], stillbirths [19,49], miscarriages [26,94], low birth weight [26,68], complicated pregnancy/delivery [36,57,58,71,74,76,90], and perinatal death [68] were reported to be associated with increased odds of PTB. Five of the included studies [36,57,58,71,74,76,90] also reported antepartum haemorrhage as a risk factor for PTB.

Parity, particularly nulliparity, and gravidity were other factors reported to be associated with PTB. Three of the included studies [48,58,80] reported nulliparity as a risk factor of PTB. However, the reports were inconsistent with two of these studies [58,80] reporting it to be associated with a higher risk of PTB while one study [48] reported a decreased risk of PTB among nuliporous women. With regards to gravidity, one study [108] reported primigravidity as being associated with less risk of PTB while another study [60] reported contrasting findings. A lack of supplementation in pregnancy was also reported to be significantly associated with an increased risk of pregnancy [18]. However, in one study [35], preconception iron supplementation was associated with a higher risk of PTB.

Multiple gestation was another consistently reported risk factor for PTB that was reported in nine of the included studies [19,27,39,47,48,50,71,77,97]. All these studies reported an increased risk of PTB with multiple gestation. Inter-pregnancy interval was also consistently reported to significantly influence PTB in 11 of the included studies [19,27,39,47,48,50,71,77,97]. In ten of these studies, a short Inter Pregnancy Interval (<24 months) was reported as being associated with a higher risk of PTB while a long inter-pregnancy interval (>60 months) was reported as a risk factor in two of these studies [61,70].

#### 3.5.3. Infections and Other Morbidities

We identified several infections and/or medical conditions in pregnancy that were reported across 56 of the included studies to be associated with PTB (Table 1). Malaria was the most commonly reported infection in pregnancy associated with PTB as reported in eleven of the included studies [27,42,46,49,70,82,84,95,99,105]. All these studies reported a higher risk of PTB with malaria infection in pregnancy. Additionally, in two studies [35,111], the seasonality patterns of PTB were observed to be paralleling with those of malaria infection, with its peak during high malaria infections. Increased odds of PTB were also reported among pregnant mothers who did not receive malaria prophylaxis during pregnancy [22] and those on malaria treatment during pregnancy [111].

Twenty-five of the included studies also reported that hypertension in pregnancy (whether chronic or gestational) was associated with increased PTB (Table 1). In one study [59], the risk of PTB varied with the severity and type of hypertension. Women with severe hypertension were at an increased risk of PTB compared to those with non-severe chronic hypertension or those with pregnancy-induced hypertension [59]. However, one study [44] reported hypertension in pregnancy as being protective against PTB (*p* = 0.001, AOR = 0.182; 95% CI = 0.067–0.493).

In 10 of the included studies [19,27,39,47,48,50,71,77,97], HIV infection in pregnancy was also reported to be associated with PTB. All these studies reported an increased risk of PTB with HIV infection. Additionally, in one study [92], increased odds of PTB were also reported among pregnant mothers with HIV and syphilis co-infection in Botswana.

Anaemia in pregnancy was also reported to be associated with increased odds of PTB in eight of the included studies [18,19,49,56,63,76,77,108], with the risk of PTB increasing with its severity [56]. Other infections or conditions in pregnancy that were reported to increase the burden of PTB were bacterial vaginosis [105], tuberculosis [84], diabetes [108], depression [78], urinary and reproductive tract infections [39,71,90,104], opportunistic infections [47], and periodontal disease [26,51,100].

#### 3.5.4. Treatment-Related Factors

Antiretroviral therapy use, time of initiation, or the type of regimen were reported in eight of the included studies [38,40,45,47,67,99,102,109] as factors associated with PTB in SSA among women living with HIV. In one study [99], antiretroviral therapy use was associated with a higher risk of PTB. In three of these studies [38,67,109], antiretroviral therapy initiated before conception was reported to be associated with a higher risk of PTB. However, in two of these studies [40,102], the risk was higher when antiretroviral treatment was initiated during pregnancy.

In terms of the type of regimen, two studies [47,102] reported a significantly higher risk of PTB when pregnant women were on protease inhibitor-based regimens. One of these studies [102] also reported a higher risk when pregnant mothers were on Nevirapine-based and Evafirenz-based regimens. However, one study [45] found a lower risk of PTB when pregnant women were on Zidovudine monotherapy. Only one study [99] reported the effect of malaria treatment on PTB. In this study, women on malaria treatment during pregnancy were more likely to have a PTB.

#### 3.5.5. Behavioural/Lifestyle Factors

Only one study [63] reported the association between drug use (alcohol consumption and smoking) in pregnancy and PTB. In this study, alcohol use and cigarette smoking were reported to be associated with an increased risk of PTB. Intimate partner violence (IPV), including physical violence, emotional violence and sexual violence, among pregnant mothers was another factor reported to increase the odds of PTB in SSA [32,94,96].

#### 3.5.6. Anthropometric Factors

Thirteen of the included studies [22,23,26,34,49,65,76,80,83,87,101,107,108] reported anthropometric measures such as body mass index (BMI), mid-upper arm circumference (MUAC), weight and height as factors associated with PTB. Four studies [23,26,101,108] assessed the effect of BMI in PTB and reported contrasting results. One study [23] reported a BMI ≥ 30 was associated with a lower risk of PTB. However, another study [26] reported that a BMI ≥ 30 and ≤ 25 was associated with a higher risk of PTB. In two of the remaining studies [101,108], a higher BMI ≥ 18.5 was associated with a lower risk of PTB.

Only one study [80] reported the association between maternal height and PTB. In this study, maternal short stature was associated with a higher risk of PTB. Only three studies [33,49,76] assessed the association between MUAC and PTB. These studies used different cut-off scores and report different results. Two of these studies [33,76] that reported MUAC being associated with a higher risk of PTB used cut-off scores of <11 and <24 cm, respectively. The remaining study [49] that reported MUAC as being associated with a lesser risk of PTB used a cut-off score of 28.6 cm.

### 3.6. Outcomes of Prematurity

In total, 24 of the included studies [9,23,39,58,112,113,114,115,116,117,118,119,120,121,122,123,124,125,126,127,128,129,130,131] reported the outcomes of babies born prematurely in SSA. We have categorized these outcomes into mortality and morbidity, and growth and development. Overall, in these studies, compared to term babies, preterm babies had a higher burden or risk of adverse outcomes.

#### 3.6.1. Mortality and Morbidity

In total, 22 of the included studies [9,23,39,58,112,113,114,115,116,117,118,119,120,121,122,124,125,126,128,129,130,131] reported preterm neonatal or infant mortality. Across these studies, the reported preterm neonatal or infant mortality ranged from 2.0 to 75.7%. Eight of these studies [9,23,39,58,117,120,127,128] compared either the risk or rates of mortality between preterm and term babies. All these studies reported higher risks or rates of mortality among preterm babies compared to their term counterparts. Across these studies, the reported morbidities among preterm babies were neonatal sepsis, neonatal jaundice, respiratory distress syndrome, asphyxia, neonatal infections, pneumonia, meningitis, pulmonary haemorrhage, respiratory problems, malnutrition and congenital malformations.

#### 3.6.2. Growth and Development

Only three studies [118,123,127] reported growth and developmental outcomes in preterm babies. One study [123] compared the incidence of neurodevelopmental delays among preterm and term infants and reported significantly higher incidence of neurodevelopmental delays among preterm infants (20.4% vs. 7.5%, *p* < 0.001). This study also compared nutritional status among these two groups and found that preterm babies were more likely to be malnourished compared with term babies [123]. However, in one study [127] that also compared neurodevelopmental outcomes between preterm and term babies, there was no significant difference in neurodevelopmental outcomes between late preterm infants and those born at term.

Only one of the included studies [118] followed up preterm babies up to two years. In this study, significantly higher proportion of preterm babies followed up to two years were moderately malnourished compared to babies born at term and there was a 0.08 increase in weight for age z score for each additional week of gestation. Additionally, in this study, more preterm babies showed developmental delays at each stage of assessment when assessed using the Malawi Developmental Assessment Tool (MDAT) [118]. They were also more likely to screen positive for neurological impairment on the Ten Question Questionnaire (TQQ) (13.9% versus 6.8%, *p* = 0.002).

## 4. Discussion

We conducted a scoping review to explore the burden of PTB, the associated risk factors, as well as its outcomes. We identified 181 eligible studies that met the inclusion criteria and were included in this review. There is a significant burden of preterm deliveries in SSA. Overall, the prevalence of PTB in SSA ranged from 3.4% to 49.4%. Differences in study contexts, methods of GA assessment, sample sizes, and study populations may explain the wide variation in the reported prevalence estimates. Regardless of these differences, the reported prevalence estimates are higher than the global estimate of 11% [4] in 75.9% of the included studies that reported prevalence in this review. These estimates are also higher than the 9% prevalence reported in Europe [132].

In this review, we observe that LMP was the most commonly used method of GA estimation. LMP is a simple method to use by all cadres of health personnel with no associated costs. This might explain its preference for use in SSA where resources are scarce in terms of finances and specialized health personnel. However, this method is more prone to errors and may be less accurate when compared to other methods such as ultrasound. Particularly, this method has been shown to underestimate gestational age in SSA [133]. This is because of the uncertainty surrounding the women’s recall of the exact date of LMP and other issues, such as delayed ovulation and every bleeding not being menstrual period [134]. Therefore, for accurate estimation of GA in SSA, and by extension, accurate determination of the burden of PTB in SSA, it is important that where possible, ultrasound is concurrently used with the other methods of estimating GA.

Several factors have been reported to predispose pregnant women to delivering a preterm baby. However, the most important risk factors for a PTB included maternal age, lack of or underutilization of ANC services, previous history of PTB, inter-pregnancy interval, premature rapture of membranes, malaria in pregnancy, hypertension in pregnancy, and HIV in pregnancy. These risk factors were reported in ten or more of the included studies to be significantly associated with PTB.

The significance of ANC is undebatable since it allows for early identification and timely management of risks in pregnancy [135,136]. However, the Sub-Saharan Africa region has one of the lowest coverage of early ANC visits in the world [137]. Early uptake of ANC and careful follow up of women with previous history of PTB and other adverse birth outcomes in previous pregnancy might prolong their pregnancy in SSA. Similarly, a timely uptake and increased number of ANC visits is vital in reducing the adverse impact of risk factors such malaria, hypertension, anaemia, and antepartum haemorrhage on the duration of pregnancy.

Age was also an important risk factor with most on the included studies reporting that both young age (<20 years) and advanced age (>35 years) in pregnancy are associated with a higher risk of PTB. Younger age being associated with a higher risk of PTB is particularly salient in SSA where about one in every five births occur among adolescents [138]. This represents a high proportion of pregnant women with a higher risk of PTB. Younger pregnant women in SSA are at risk of PTB because during pregnancy, they are less likely to attend ANC due to fears of HIV testing, financial barriers and lack of knowledge on ANC [62,139]. Therefore, efforts are needed to scale up the uptake of ANC among this vulnerable group. Older pregnant women are also at more risk of PTB because they are more likely to have co-morbid conditions such as hypertension, diabetes, and obesity, which are risk factors for PTB [140].

Infections and morbidities in pregnancy such as malaria, HIV and hypertension were also identified as important risk factors associated with PTB in SSA. This calls for early identification and management of these conditions during the antenatal period. Therefore, there is a need for pregnant women to attend antenatal clinics early in pregnancy as well as attend all the recommended visits of at least four visits during the entire pregnancy so they can receive targeted care and management of these conditions during the entire period.

Among the 12 studies that reported an association between inter-pregnancy interval and PTB, ten of these studies reported an association between a shorter inter-pregnancy interval and PTB. This association has also been observed in a large cohort study involving over 150,000 women in United states [141]. A plausible explanation for this observation has been suggested in the “maternal depletion hypothesis” where it has been thought that shorter inter-pregnancy intervals do not allow sufficient time for women to recover from the physiological strain of the previous pregnancy [142,143]. This highlights the importance of planned pregnancy as an important component of women’s health and strategy to reduce PTB. However, two other studies [61,70] reported that a longer inter-pregnancy interval was associated with a higher risk of PTB. Therefore, more studies are needed in SSA to explore these associations and to provide clearer insights.

In this review, mothers with a previous history of PTB were also more likely to have PTB. This risk increased further with an increase in the number of previous preterm births. This is consistent with findings from a previous systematic review that assessed the risk of spontaneous recurrent preterm births [144]. Previous PTBs may predict recurrent PTBs because the underlying causes and predisposing factors in the initial PTB might be similar in the subsequent pregnancies [144]. Finally, the premature rupture of membranes (PROM) was also an important risk factor for PTB in SSA. PROM has been consistently associated with PTB [145] and has been reported to be responsible for about 40% of preterm births [146]. Identifying these women as high risk for subsequent preterm deliveries and increasing monitoring in future pregnancies is important.

Immediate and long-term outcomes of PTB are generally poor compared to term babies. In this review, preterm babies had higher risk or rates of mortality and morbidity compared to term babies. They also showed poorer growth and development. These outcomes could be improved but this is dependent on the quality of hospital care provided in the critical neonatal period. However, the majority of hospitals in SSA lack advanced care. Nonetheless, the uptake of simple and affordable neonatal care practices such as Kangaroo mother care, delayed birthing, and hygienic newborn umbilical cord care have been shown to be effective in improving outcomes but their implementation and coverage has been variable [147]. Therefore, promoting optimal neonatal care practices for both facility and home births can potentially save thousands of preterm neonatal deaths in SSA [7]. For those born extremely premature, they face the highest risk of dying following admission due to lack of technologies to provide advanced care. Thus, the majority (>90%) of extreme babies in SSA die within the first week of admission [130] while the majority (>90%) of their counterparts in high-income regions of world survive hospital admission [7].

From this review, there is paucity of data from SSA on the long-term developmental outcomes among children who were born preterm. We found very few studies that assessed mortality/morbidity, growth and development patterns among preterm babies who survived infancy. None of the included studies have assessed the long-term impact of prematurity on the different domains of neurodevelopment including brain development, motor skills, cognition, language development and school performance among pre-school, school going children or adolescents. This seems to be an important gap that needs to be addressed. In particular, strategies that are workable in low-income countries are needed. It is unlikely that formal neurodevelopmental assessments that are conducted in a hospital clinical environment in tertiary hospitals in high-income settings will be possible in low-income settings. Other methods of assessment and data capture such as parent questionnaires and app-based home assessment need to be researched.

### Limitations

Our study had some limitations. We could not retrieve all the articles identified in the search as we only included studies published in English language. There is a possibility that we might have excluded some articles published in other languages. Also, as with scoping reviews, we could not assess the quality and rigor of the included studies.

## 5. Conclusions

There is a significant burden of PTB in SSA, and the burden is very likely to be underestimated, given that LMP, a less accurate measure of GA, has been the most commonly used method of GA estimation in SSA. Although several risk factors have been reported to predispose pregnant women to preterm delivery, several questions remain un-answered. However, a multi-factorial approach will greatly reduce the burden of PTB. Additionally, there is a need for collaborative research networks, cooperation between governments and non-governmental organizations, a standardized approach to data collection, and the empowerment of communities by disseminating knowledge.

## Figures and Tables

**Figure 1 ijerph-19-10537-f001:**
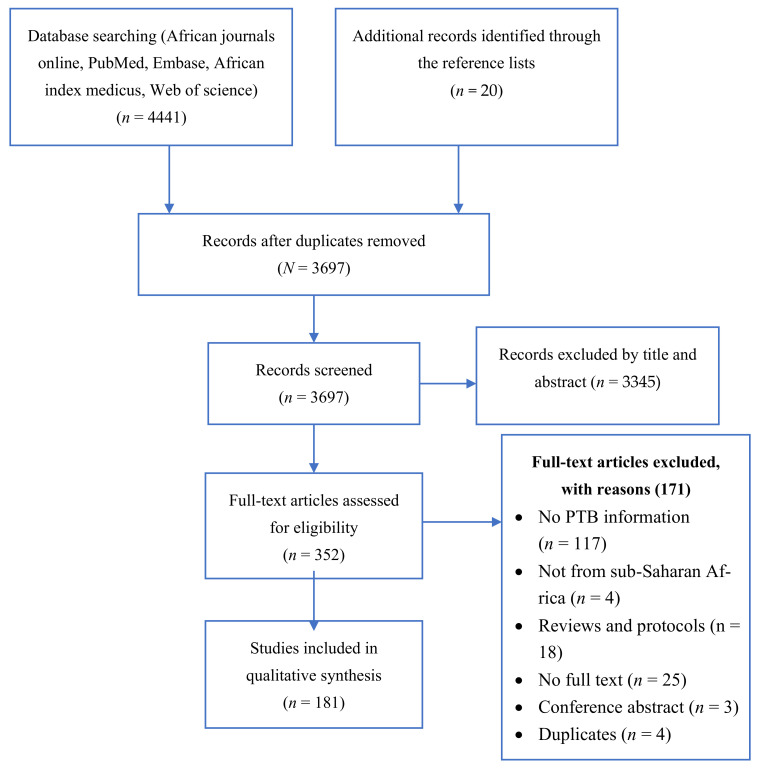
Flow diagram showing the study selection process.

**Figure 2 ijerph-19-10537-f002:**
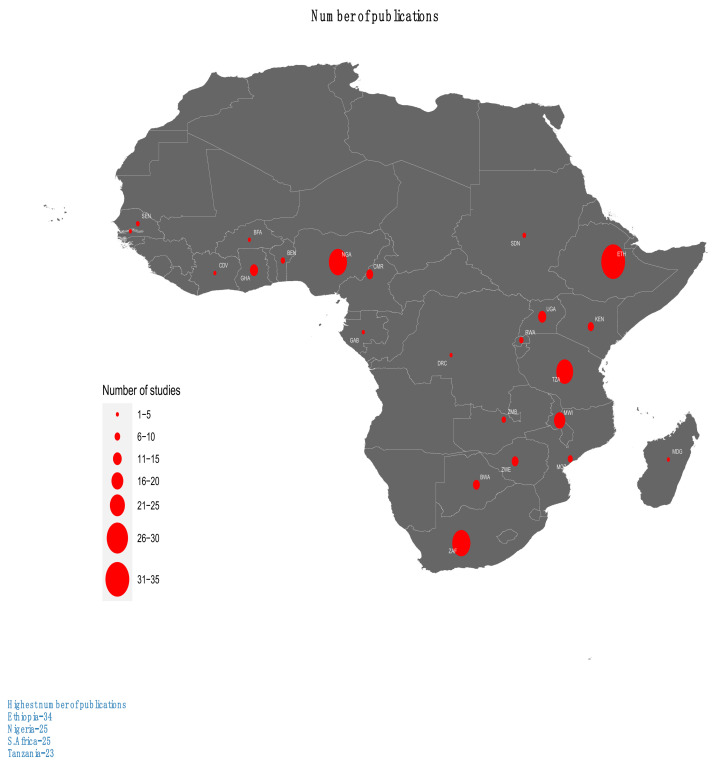
Geographical distribution of the included studies.

**Table 1 ijerph-19-10537-t001:** Risk factors of PTB in SSA.

First Author	Measure of Effect (Precision)	Sociodemographic Factors	Obstetric Factors	Infections and/or Morbidities	Lifestyle or Behavioral Factors	Anthropometric Factors	Treatment-Related Factors	Factors Adjusted for
Abadiga et al., 2021 [18]	AOR (95% CI)	NR	<2 ANC visits (6.0, 2.7–13.5), PROM (3.6, 1.8–7.1), IPI < 24 months (3.0, 1.8–7.1), lack of supplementation during pregnancy (2.4, 1.5–3.9)	PIH (3.1, 1.8–5.5), anemia during pregnancy (4.2, 2.1–8.3)	NR	NR	NR	Age at marriage, ethnicity, age of mother, husband and mothers occupation, residence, monthly income, time to reach a health facility, mode of delivery, plan of pregnancy, history of abortion, history of DM and hypertension, HIV/AIDS status, anemia during pregnancy, sexually transmitted disease, substance use, parity, the experience of stress, and maternal MUAC
Abaraya et al., 2018 [19]	AOR (95% CI)	NR	Previous PTB (6.4, 2.9–13.9), PROM (11.7, 6.2–21.9), <4ANC (4.8, 1.2–19.0), IPI < 24 months (2.7, 1.7–4.5), multiple pregnancies (4.5, 2.4–8.5), previous stillbirth (4.2, 2.0–8.9), APH (8.1, 3.5–18.5)	Preeclampsia (6.6, 3.3–13.4), Anemia (2.7, 1.0–7.01), PIH (6.0, 2.3–15.6)	NR	NR	NR	Number of ANC visits, hemoglobin levels, PIH, PROM, cardiac disease
Abdo et al., 2020 [20]			PROM (4.2, 1.4–12.9)	Preeclampsia (5.1, 2.0–13.3)				Age, residence, previous PTB, parity, ANC visits
Abebe et al., 2020 [21]	AOR (95% CI)	Adolescence (2.9, 1.5–5.5)	APH (4.9, 1.9–12.9)	PIH (3.0, 1.4–6.6)	NR	NR	NR	Residence
Abrams et al., 2004 [22]	AOR (95% CI)		<4 ANC visits (2.6, 1.4–4.8)	HIV infection (5.0, 1.7–14.6)	NR	Weight ≤ 50 kg (2.1, 1.1–4.1)	Not on Malaria prophylaxis (3.3, 1.3–7.9)	Gravidity, age, weight, Anemia, folate supplementation, malaria prophylaxis, malaria infection, CD4 count, placental malaria, syphilis
Adane et al., 2014 [8]	AOR (95% CI)	NR	Previous poor birth outcome (PTB/LBW) (3.10, 1.20–8.36)	NR	NR	NR	NR	Hypertension
Adu-Bonsaffoh et al., 2019 [23]	AOR (95% CI)	Age ≥ 35 years (1.5, 1.1–1.9),	<4 ANC visits (5.1, 4.1–6.6), PROM (3.1, 2.2–4.6)	Hypertension (3.3, 2.6–4.2)	NR	BMI ≥ 30 (0.6, 0.5–0.8),	NR	Age, parity, marital status, education, BMI, ANC visits, PROM
Agbla et al., 2006 [24]	AOR (95% CI)	Carrying heavy loads > 5 days/week (5.0, 1.4–18.8), walking and carrying heavy loads > 5 days/week (6.9, 1.5–32.2)	NR	NR	NR	NR	NR	Age, education, health problems during pregnancy, twin birth
Akintayo et al., 2015 [9]	AOR (95% CI)	Age > 35 years (2.16; 1.36–3.42),	Previous PTB (6.4, 2.5–16.6), unbooked for ANC care (2.5, 1.8–3.6), PROM (11.1, 6.6–18.6), APH (10.9, 4.9–24.1), (32.2, 13.1–79.5)	PIH (6.4, 3.8–10.9)	NR	NR	NR	Parity, socioeconomic status, education, fetal sex, previous history of surgical abortion
Akintije et al., 2020 [25]	AOR (95% CI)	NR	ANC visits ≥ 4 (0.22, 0.1–0.5)	NR	NR	NR	NR	Age, parity, marital status, previous cesarean section, type of pregnancy, mode of delivery, living area, sex of the newborn
Alhaj et al., 2010 [26]	AOR (95% CI)	NR	Previous PTB (3.2, 1.0–9.8), previous miscarriage (2.1, 1.0–4.5), previous CS (5.4, 1.7–17.3), IPI < 18 months (2.0, 1.3–5.4), Vaginal bleeding (1.3–18.1)	Dental maneuvers (3.7, 1.1–11.8)	NR	BMI < 25 (3.0, 1.0–8.3), BMI > 30 (3.1, 1.0–9.0)	NR	History of malaria, miscarriage and PTB, CS delivery, diabetes, hypertension, vaginal bleeding, APH, placenta praevia, dental procedures, BMI, sex of the newborn
Aregawi et al., 2019 [27]	AOR (95% CI)	Rural residence (2.1, 1.1–4.2)	IPI < 24 months (5.4, 1.3–22.1), Previous PTB (3.7, 1.1–16.3), PROM (4.1, 1.9–8.9), induced labor (2.5, 1.1–5.9), multiple pregnancy (5.7, 2.3–14.3)	Malaria during pregnancy (4.7, 2.0–11.2), presence of a chronic illness (4.6, 1.8–11.3)	NR	NR	NR	NR
Asefa et al., 2020 [28]	AOR (95% CI)	Age ≥ 35 (3.0, 1.4–6.3)	NR	NR	NR	NR	NR	Residence, ANC visits, gravidity
Aseidu et al., 2019 [29]	AOR (95% CI)	NR	PROM (2.3, 1.0–5.5), ANC > 4 (0.2, 0.1–0.4)	Preeclampsia/eclampsia (3.4, 1.0–11.9)	NR	NR	NR	Parity, Number of ANC visits, APH before, multiple birth, Mode of delivery, (P)PROM labor, Gestational Diabetes Mellitus (GDM), PIH, Pre-eclampsia/eclampsia, Preterm history, Birth Interval, Previous mode of delivery
Bater et al., 2020 [30]	AOR (95% CI)	Residence (0.6, 0.5–0.8), age > 20 years (0.8, 0.6–0.9), single (1.4, 1.0–2.0), home delivery (1.3, 1.1–1.5)	IPI > 24 months (0.8, 0.6–0.9), ANC > 4 (0.6, 0.5–0.7)	NR	NR	NR	NR	Residence, being age, education, diet, IPI, ANC visits, household food insecurity, not taking deworming medication or iron tablets during pregnancy, and delivering at home
Berhane et al., 2019 [31]	AOR (95% CI)	Unmarried mothers (4.1, 1.2–14.1)	NR	NR	NR	NR	NR	Sex of the newborn, marital status, maternal education, socioeconomic index, gravidity, parity
Berhanie et al., 2019 [32]	AOR (95% CI)	NR	NR	NR	IPV (2.5, 2.2–2.9), Physical violence (5.3, 4.0–7.1)	NR	NR	IPV, sexual violence, physical violence, psychological violence, and controlling behaviors
Berhe et al., 2019 [33]	ARR (95% CI)	NR	NR	PIH (5.1, 3.4–8.0)	NR	NR	NR	Maternal age, wealth status, educational status, residence, gravidity, mode of delivery, anemia and maternal undernutrition
Berhe et al., 2019 [34]	AOR (95% CI)	Unmarried mothers (5.2, 1.8–15.1)	NR	NR	NR	MUAC < 11 cm (2.4, 1.2–4.9)	NR	Age, income, education, occupation, height, pregnancy status, diet, medical problems
Brabin et al., 2019 [35]	ARR (99% CI)	NR	Préconception iron supplementation (2.2, 1.4–3.6)	NR	NR	NR	NR	MUAC at baseline, bed net use, birth month
Brhane et al., 2019 [36]	AHR (95% CI)	NR	IPI < 24 months (6.9, 3.1–15.3), ANC < 4 (2.2, 1.0–4.9), pregnancy complication (3.2, 1.6–6.3) birth defect (8.0, 2.6–25.1)	NR	NR	NR	NR	Maternal residence, planned pregnancy, history of perinatal death and maternal hemoglobin level
Butali et al., 2016 [37]	AOR (95% CI)	Age > 35 years (1.4, 1.1–1.7)	PMTCT attendance (0.9, 0.6–1.3), PROM (4.03, 2.5–6.40), unbooked for ANC (2.15, 1.6–2.9)	Hypertension (2.8, 1.9–4.0)	NR	NR	NR	Maternal age, parity, fetal position, delivery method and booking status
Chen et al., 2012 [38]	AOR (95% CI)	NR	NR	HIV infection (1.3, 1.2–1.4)	NR	NR	Preconception HAART (1.2, 1.1–1.4)	NR
Chiabi et al., 2013 [39]	AOR (95% CI)	Student mother (0.4, 0.2–0.9), married (0.4, 0.19–0.8)	Multiple gestation (3.8, 2.6–5.4), congenital malformation (2.8, 1.2–6.2), ≥4 ANC visits (0.2, 0.1–0.4)	UTI (39.0, 17.2–88.6)	NR	NR	NR	Gender, age, occupation, education, marital status, residence, parity, gravidity, ANC visits, place of ANC visit
Conroy et al., 2017 [40]	AOR (95% CI)	NR	NR	NR	NR	NR	First trimester ART initiation (1.31, 1.03–1.68)	NR
Dadi et al., 2020 [41]	ARR (95% CI)	Muslim (1.61, 1.17–2.22), government employee (1.49, 1.00–2.19), have fear of delivery (1.46, 1.06–2.01)	No ANC uptake (1.77, 1.03–3.03)	NR	NR	NR	NR	MUAC, depressive symptoms, coping with stress
De Beaudrap et al., 2013 [42]	AOR (95% CI)	NR	NR	Malaria (1.9, 1.1– 3.5), HIV (2.33, 1.17–4.64	NR	NR	NR	Maternal age, education level, residence, HIV status, number of clinic follow-up visits, newborn’s sex
Debelo et al., 2020 [43]	ARR (95% CI)	Age > 35 (2.7, 1.8–3.8)	NR	NR	NR	NR	NR	Number of alive child(ren), wealth quintile, husband occupational status, gravidity, husband educational status, maternal educational status, parity, ANC follow-up, previous adverse pregnancy outcomes, health insurance, place of residence
Deressa, 2018, Ethiopia [44]	AOR (95% CI)	NR	NR	PIH (0.182, 0.067–0.493), HIV (3.4, 1.048–11.1)	NR	NR	NR	NR
Ejigu et al., 2019 [45]	AOR (95% CI)	NR	NR	NR	NR	NR	Zidovudine monotherapy (0.4, 0.2–0.6)	Maternal age, weight, marital status, education, parity, CD4 cell count during pregnancy and WHO clinical stage during pregnancy
Elphinstone et al., 2019 [46]	ARR (95% CI)	NR	NR	Malaria positive before 24 weeks (1.7, 1.2–2.3)	NR	NR	NR	Treatment arm (ISTp versus IPTp), maternal age, gravidity, socioeconomic status, education status, body mass index (BMI), and hemoglobin at Visit 1
Ezechi et al., 2012 [47]	AOR (95% CI)	NR	Multiple pregnancy (8.6, 6.7–12.9)	Presence of opportunistic infection (1.9, 1.1–5.7)	NR	NR	PI based regimen (5.4, 3.4–7.8)	Birth weight, stage of HIV disease, reproductive tract infection and medical disorders
Feresu et al., 2010 [48]	APR (95% CI)	Rural residence (1.2, 1.1–1.3)	Multiple gestation (4.2, 3.7–4.6), ANC < 1 visit (3.0, 2.8–3.3), nulliparity (0.90, 0.83–0.99)	NR	NR	NR	NR	Age, sex, residence antenatal care, delivery type and parity
Feresu et al., 2004 [49]	ARR (95% CI)	NR	History of abortion/stillbirth (1.5, 1.1–2.1), APH (3.1, 1.9–5.0), placental Previa (3.3, 1.3–8.1)	Eclampsia (3.6, 1.7–7.6), anemia (4.1, 1.8–9.4), Malaria (2.9, 1.7–5.0)	NR	MUAC < 28.6 cm (0.95, 0.92–0.99),	NR	Maternal age, antenatal care attendance, referral status, drinking home brew during pregnancy, and history of abortion or stillbirth
Gejo et al., 2021 [50]	AOR (95% CI)	Urban residence (0.5, 0.2–1.0)	No ANC (0.1, 0.0–0.7), PROM (3.8, 1.5–9.7), multiple pregnancies (5.5, 2.5–12.4)	PIH (3.8, 1.4–10.1)	NR	NR	NR	Parity, residency, history of abortion, history of PTB, history of stillbirth, UTI, diabetes mellitus, anemia, ANC follow up, labour, APH, PROM, PIH, polyhydramnios and multiple pregnancy
Gesase et al., 2018 [51]	AOR (95% CI)	NR	NR	Periodontal disease (2.3, 1.3–4.3)	NR	NR	NR	Age, parity, previous preterm birth, he same adverse fetal outcome, and pre-eclampsia
Gumede et al., 2017 [52]	AOR (95% CI)	NR	ANC < 1 visit (1.6, 1.4–1.8)	HIV infection (1.3, 1.2–1.4)	NR	NR	NR	HIV status, study site, infant sex
Habib et al., 2011 [53]	ARR (95% CI)	Unknown HIV status (1.4, 1.2–1.6)		HIV infection (1.8, 1.1–2.7)				NR
Hassen et al., 2021 [54]	AOR (95% CI)	NR	Previous PTB (3.5, 1.4–8.8), IPI < 24 months (4.5, 2.0–10.2), history of obstetric complications (3.8, 1.6–9.0)	NR	NR	NR	NR	Monthly family income, history of PTB, IPI, ANC visit, mode of delivery in current pregnancy, experiencing obstetric complication in current pregnancy, maternal weight, infant birth weight, and presence of birth asphyxia
Hussain et al., 2011 [55]	AOR (95% CI)	NR	NR	HIV infection (1.4, 1.1–1.7)	NR	NR	NR	NR
Hussein et al., 2009 [56]	AOR (95% CI)	NR	NR	Mild anemia (1.4, 1.1–1.9), severe anemia (4.1, 2.5–6.6)	NR	NR	NR	NR
Hussein et al., 2020 [57]	ARR (95% CI)	Charcoal use (1.5, 1.1–2.1)	NR	NR	NR	NR	NR	Maternal malaria at birth, parity and number of cooking sessions a day
Iyoke et al., 2015 [58]	AOR (95% CI)	Unmarried mothers (2.4, 1.5–3.7)	ANC < 1 visit (2.6, 1.9–6.1), previous PTB (5.1, 2.7–9.1), pregnancy complication (5.1, 2.4–11.8), nulliparity (2.1, 1.2–4.9)	NR	NR	NR	NR	Educational status, occupation, PTB, parity, marital status and the presence of complicationsof pregnancy (antepartum hemorrhage, preeclampsia/eclampsia or PROM)
Johnson et al., 2016 [59]	ARR (95% CI)	NR	NR	Gestational hypertension (1.2, 1.1–14), chronic hypertension (2.3, 2.1–2.6), severe hypertension (4.4, 3.2–6.2)	NR	NR	NR	Maternal age, marital status, salaried employment, and booking weigh
Kalanda et al., 2006 [60]	AOR (95% CI)	NR	Primigravida (2.3, 1.3–4.0), <5 ANC visits (2.2, 1.3–3.7)	NR	NR	NR	NR	Maternal age, ANC visits, maternal height, MUACHb leval at recruitment, peripheral or placental and peripheral malaria, taking ferrous sulphate supplements and taking sulphadoxine–pyrimethamine
Kalengo et al., 2020 [61]	ARR (95% CI)	NR	Previous PTB (1.9, 1.5–2.3), IPI > 59 months (1.4, 1.0–1.9)	Preeclampsia (1.5, 1.1–2.0)	NR	NR	NR	Mother’s age, mother’s education, PROM and alcohol use
Kassa et al., 2019 [62]	AOR (95% CI)	Age < 20 (1.7, 1.1–2.5)	ANC ≥ 1 visit (0.4, 0.2–0.8)	Preeclampsia (2.6, 1.2–5.5)	NR	NR	NR	Maternal age, residence, school attendance, marital status, wealth status, educational status of the father and the mother, anemia, iron-folic acid supplementation during current pregnancy, ANC attendance, partner involvement in ANC, experience of at least one form of gender based violence (physical, sexual or psychological violence) during the current pregnancy, and preeclampsia
Kelkay et al., 2018 [63]	AOR (95% CI)	NR	History of abortion (2.3, 1.2–4.9)	Anemia (2.4, 1.1–5.2)	Cigarette/alcohol use (3.6, 1.6–8.2)	NR	NR	Parity, medication intake, smoking cigarette/drinking alcohol during the most recent pregnancy, history of abortion, hemoglobin level, a physical congenital defect in the most recent baby, and history of bearing low birth weight baby, history of still birth, history of PTB, malaria
Kongnyuy et al., 2008 [64]	AOR (95% CI)	Age < 20 (1.8, 1.2–2.5)	NR	NR	NR	NR	NR	Gravidity, antenatal visits, marital status, employment status and level of education
Koss et al., 2014 [65]	AOR (95% CI)	NR	NR	NR	NR	Weight gain < 0.1 Kg/week (2.4, 1.2–4.4)	NR	Adjustment for time since HIV diagnosis and ART regimen
Kumwenda et al., 2017 [66]	AOR (95% CI)	Age < 20 (2.6, 1.2–5.8)	NR	NR	NR	NR	NR	NR
Li et al., 2016 [67]	ARR (95% CI)	NR	NR	Hypertension (1.3, 1.1–1.5)	NR	NR	Preconception HAART use (1.2, 1.1–1.5)	CD4+ cell count, maternal nutritional status
Mahande et al., 2013 [68]	ARR (95% CI)	NR	Previous PTB (2.7, 2.1–3.4), previous perinatal death (2.6, 1.9–3.5), previous LBW (2.9, 2.3–3.6)	Previous preeclampsia (2.5, 1.7–3.7)	NR	NR	NR	Maternal age and maternal education
Mahande et al., 2016 [69]	AOR (95% CI)	NR	IPI < 24 months (1.5, 1.3–1.7), IPI ≥ 60 months (1.1, 1.02–1.2)	NR	NR	NR	NR	Maternal age, maternal marital status, maternal educational status, maternal occupation, parity, area of residence, number of antenatal care visits (ANC), use of family planning methods and use of alcohol during pregnancy
Mahande et al., 2016 [70]	AOR (95% CI)	NR	NR	Malaria (1.12, 1.01–1·26), amebiasis (1.8, 1.1–2.9)	NR	NR	NR	Maternal age, parity, antenatal care visits maternal education, maternal occupation, area of residence, marital status
Mahapula et al., 2016 [71]	AOR (95% CI)	NR	No ANC (5.1, 1.4–17.8), vaginal discharge (5.2, 1.1–24.4), public perinatal care (2.1, 1.1–4.1), multiple pregnancy (8.6, 4.5–16.5), complications (2.7, 1.3–5.3), cervical incompetency (11.6, 1.1–121·5), polyhydramnios (8.3, 1.7–40.2)	Untreated UTI (2.7, 1.2–6.1)	NR	NR	NR	NR
Malaba et al., 2017 [72]	AOR (95% CI)	NR	NR	HIV infection (1.9, 1.3–2.8)	NR	NR	NR	Age, parity, BMI and previous PTB
Malaba et al., 2020 [73]	ARR (95% CI)	NR	NR	PIH (1.5, 1.1–2.0)	NR	NR	NR	Age, parity, BMI and previous PTB
Mboya et al., 2021 [74]	AOR (95% CI)	NR	Referred for delivery (1.3, 1.1–1.5), <4 ANC visits (5.6, 4.7–6.8), PROM (1.6, 1.1–2.5), placental Previa (8.1, 3.6–18.1)	Preeclampsia/eclampsia (1.6, 1.3–2.0),	NR	NR	NR	Maternal age, level of education, referral status, pre-eclampsia/eclampsia, number of ANC visits, parity, PROM, abruption placenta, placenta previa, delivery mode, child’s birth weight, perinatal status and year of birth.
Mehari et al., 2020 [75]	AOR (95% CI)	Age ≥ 35 years (3.6, 1.5–8.9)	NR	NR	NR	NR	NR	Residence, number of ANC visits, malpresentation, gravidity, bad obstetric history, Pregnancy Induced Hypertension, APH, PROM and Amniotic Fluids disorders
Mekonen et al., 2019 [76]	AOR (95% CI)	NR	Obstetric complications (6.6, 3.4–12.6), <4 ANC visits (5.1, 1.7–15.4), PROM (3.0, 1.5–6.2)	Anemia (2.9, 1.3–6.6)	NR	MUAC < 24 cm (2.6, 1.1–6.1)	NR	NR
Mekuriyaw et al., 2020 [77]	AOR (95% CI)	No education (2.2, 1.3–3.9)	PROM (6.4, 3.2–12.8), multiple pregnancy (4.1, 1.7–9.8), previous abortion (2.9, 1.3–6.4)	Anemia (2.8, 1.1–7.3), PIH (4.7, 2.5–9.0),	NR	NR	NR	NR
Mochache et al., 2018 [78]	ARR (95% CI)	NR	NR	Depression (3.6, 1.7–7.5)	NR	NR	NR	NR
Mombo-Ngoma et al., 2016 [11]	AOR (95% CI)	Age ≤ 16 years (2.16, 1.10–4.24)	NR	NR	NR	NR	NR	Country, first antenatal clinic visit, treatment group and infant sex
Moodley et al., 2017 [79]	AOR (95% CI)	Age ≥ 35 years (0.4 (0.2–0·90)	NR	NR	NR	NR	NR	Age, gravidity, socioeconomic status, HIV infection, chlamydia, trichomonas, gonorrhea
Muhihi et al., 2016 [80]	ARR (95% CI)	Age < 20 years (1.2, 1.1–1.4)	Nulliparity (1.2, 1.1–1.3)	NR	NR	Mothers’ height < 150 cm (1.3, 1.1–1.7)	NR	NR
Muhumed, 2021, Ethiopia [81]	AOR (95% CI)	Rural residence (4.5, 1.4–14.4)	History of abortion (5.0, 1.9–13.5)	Hypertension (3.2, 1.1–10.2)	NR	NR	NR	NR
N’Dao et al., 2006 [82]	AOR (95% OR)	NR	NR	Placental malaria (3.5, 1.8–6.7)	NR	NR	NR	NR
Ngandu et al., 2021 [83]	AOR (95% CI)	Secondary level education (2·21, 1.07–4.59), South African (3.72 (1.51–9.15)	Duration between first prenatal visit and delivery (0.72, 0.68–0.77), vaginal delivery (1.97, 1.22–3.15), GA determined at first prenatal visit (0.93, 0.89–0.98),	NR	NR	Weight at delivery (0.99, 0.97–1.00), gestational weight gain per week (0.21, 0.07–0.61)	NR	Newborn and maternal characteristics, and country
Noble et al., 2005 [84]	AOR (95% CI)	NR	NR	Tuberculosis (10.2, 1.2–89.9), Malaria (2.4, 1.1–5.3)	NR	NR	NR	Maternal age, gravidity, education, marital status, pre-pregnancy body mass index, and maternal use of and iron supplements during the current pregnancy
Oluwole et al., 2020 [85]	AOR (95% CI)	NR	NR	Vitamin D serum level < 30 ng/mL (9.4, 2.4–36.5)	NR	NR	NR	Age, level of education, religion, booking status, alcohol intake, smoking of tobacco, and skin color
Olusanya et al., 2010 [86]	AOR (95% CI)	Unmarried (1.7, 1.1–2.7), shared sanitation (1.3, 1.1–1.5), small trading (1.5, 1.3–1.9), fulltime job (1.3, 1.1–1.6)	PROM (3.6, 2.0–6.5), APH (3.5, 2.0–6.4), No history of CS (1.4, 1.1–1.9), No ANC (1.3, 1.1–1.5)	Hypertension (2.2, 1.7–2.9), Maternal disease (1.5, 1.1–2.1),	NR	NR	NR	Maternal and infant perinatal factors
Omokhodion et al., 2010 [12]	AOR (95% CI)	Exposure to vibration at work (2.40, 1.21–4.77)	Nulliparity (2.24, 1.26–3.97), history of PTB (6.45, 1.41–29.53), ≤ 4 ANC visits (4.05, 1.70–9.66), prolonged PROM (6.41, 1.86–22.11)	NR	NR	NR	NR	Age, marital status, matenal education, type of residence, cooking with kerosense, parity, history of PTB, prolonged PROM, hospital admission during preganancy, ANC visits, hypertension
Osman et al., 2001 [87]	AOR (95% CI)	NR	NR	NR	NR	Weight < 49 Kg (3.0, 1.7–5.2), Weight gain < 1 Kg (2.8, 1.6–5.0)	NR	Number of children alive
Padonou et al., 2014 [88]	AOR (95% CI)	NR	NR	NR	NR	NR	NR	Parity, maternal anaemia, presence of placental malaria, infections
Parek et al., 2011 [89]	AOR (95% CI)	NR	NR	HIV infection (1.7, 1.3–2.2), Hypertension (1.8, 1.2–2.6)	NR	NR	NR	Maternal age, marital status, employment status, time of Initiation of prenatal care
Regasa et al., 2021 [90]	AOR (95% CI)	NR	No ANC (3.2, 1.4–7.4), 1–2 ANC visits (2.3, 1.2–4.4), Previous PTB (5.2, 2.3–20.9), IPI < 24 months (4.4, 2.1–9.5), obstetric complication (2.5, 1.3–4.7)	Reproductive tract infection (2.5, 1.0–6.3)	NR	NR	NR	History of abortion, male sex, lack of ANC follow up, iron supplementation, anxiety during pregnancy
Rugumisa et al., 2020 [91]	AOR (95% CI)	NR	Previous PTB (13.2, 1.7–102.0), placenta Previa (12.6, 1.6–98.0), PROM (8.8, 1.3–46.0)	NR	NR	NR	NR	NR
Shava et al., 2019 [92]	AOR (95% CI)			HIV and Syphilis coinfections (1.5, 1.1–2.1)				Maternal age, marital status, occupation, education, parity, low maternal haemoglobin
Siakwa et al., 2020 [93]	AOR (95% CI)	Age 20–34 (1.6, 1.4–2.6)	NR	NR	NR	NR	NR	NR
Sigalla et al., 2017 [94]	AOR (95% CI)	NR	Previous miscarriage (2.2, 1.1–4.1)	NR	Physical violence (2.9, 1.3–6.5)	NR	NR	Previous PTB, women’s age, education level, occupation, and alcohol consumption, previous LBW
Sullivan et al., 1999 [95]	AOR (95% CI)	Delivery in rainy season (3.9, 1.8–8.8), no education (3.5, 1.6–7.8)	NR	Malaria (3.3, 1.3–8.8)	NR	NR	NR	Placental parasitemia, maternal peripheral parasitemia- delivery, ANC visits, placental weight
Tadese et al., 2020 [96]	AOR (95% CI)	NR	NR	NR	IPV (2.9, 1.4–6.2), Physical IPV (2.6, 1.3–6.8), emotional IPV (3.1, 1.4–6.9)	NR	NR	Residence, education level, maternal age, ANC visits, previous history of adverse birth outcomes, current maternal and husband substance use during pregnancy, medical problems during pregnancy, any other IPV
Teklay et al., 2018 [97]	AOR (95% CI)	NR	<4 ANC visits (2.2, 1.2–3.9), multiple pregnancy (2.5, 1.1–5.3), fetal distress (4.0, 1.9–8.2), birth defect (3.2, 1.2–83)	Hypertension (3.2, 1.6–6.7)	NR	NR	NR	Maternal height, PROM
Tembo et al., 2020 [98]	AOR (95% CI)	Age < 20 (1.4, 1.1–1.8)	NR	NR	NR	NR	NR	NR
Ticconi et al., 2003 [99]	AOR (95% CI)	NR	NR	HIV infection (4.1, 2.1–7.8), malaria (25.5, 12.2–52.9)	NR	NR	HIV treatment (9.1, 4.0–20.8), Malaria treatment (23.2, 11.4–47.2)	Age, parity, malaria, malaria treatment, HIV treatment
Uwambaye et al., 2021 [100]	AOR (95% CI)	NR	NR	Periodontitis (6.4, 3.9–10.4)	NR	NR	NR	Employment status, tobacco use, history of Malaria, History of UTI, History of stress during pregnancy, other causes of stress
Van Den Broek et al., 2014 [101]	AOR (95% CI)	NR	Previous PTB (2.1, 1.2–3.80)	NR	NR	BMI > 18·5 (0.91, 0.85–0.97), Weight gain (0.89, 0.82–0.97) ·	NR	
Van der Merwe et al., 2011 [102]	AOR (95% CI)	NR	NR	NR	NR	NR	HAART exposure < 28 weeks; PI-based (3.0, 1.1–8.4), NVP-based (5.4, 2.1–13.7), EFV-based (5.6, 2.1–15.2)	Age, anaemia, malaria, BMI, previous neonataL death or stillbirth
Wagura et al., 2018 [103]	AOR (95% CI)	NR	APH (4.3, 1.5–12.0), Prolonged PROM (5.3, 2.3–12.2)	PIH (7.8, 3.7–16.5)	NR	NR	NR	Maternal age, parity, twin gestation, UTI, previous preterm birth
Wakeyo et al., 2020 [104]	AOR (95% CI)	Secondary education and above (0.1, 0.1–0.7),	ANC (0.4, 0.2–0.9), history of abortion (2.3, 1.1–5.0), Previous PTB (5.0, 1.6–15.0)	UTI (3.6, 1.1–11.0)	NR	NR	NR	HIV status, employment status
Watson-Jones et al., 2007 [105]	AOR (95% CI)	NR	NR	Malaria (3.2, 1.9–5.2), Treated Bacterial Vaginosis (0.9, 0.6–1.4), untreated Bacterial Vaginosis (3.0, 1.3–6.6)	NR	NR	NR	Age, occupation, gravidity, bacterial vaginosis during pregnancy, HIV at delivery and maternal malaria
Woday et al., 2019 [106]	AOR (95% CI)	Age < 24 years (3.5, 1.1–10.8), rural residence (3.0, 1.2–7.5), No education (4.6, 1.1–8.6)	IPI < 24 months (2.5, 1.1–5.8), No ANC (10.8, 4.4–26.3), Previous adverse birth outcomes (3.5, 1.5–8.0)	Medical problems in pregnancy (13.9, 4.4–24.23)	NR	NR	NR	MUAC, HTN status in pregnancy, Alcohol consumption in pregnancy
Young et al., 2012 [107]	AOR (95% CI)	NR	NR	NR	NR	Gestational weight gain < 0·1 kg/week (3.5, 1.2–10.1)	NR	Birth spacing < 2 years, Baseline CD4 count, Maternal weight at 7 months gestation, socioeconomic status
Zar 2020 et al., [108]	Adjusted B (95% CI)	Mothers who were food insecure (−0.542), moderate-high (0.543) or high (0.605) socioeconomic status	Primigravida (0.481)	PIH (−1.226), preeclampsia or eclampsia (1.741), gestational diabetes (−2.837), anemia (0.420)	NR	Higher BMI (0·078)	NR	Substance abuse, mental health and psychosocial factors, HIV infection, maternal age, ANC clinic attended, marital status
Zash et al., 2018 [109]	ARR (95% CI)	NR	NR	NR	NR	NR	Preconception ART (1.3, 1.1–1.7)	NR
Zewde et al., 2020 [110]	AOR (95% CI)	NR	Previous PTB (2.6, 1.3–5.2)	NR	NR	NR	NR	ANC visits, history of APH, hemoglobin levels, birth interval less than 24 months, history of chronic disease

ANC—Antenatal Care, APH—Antepartum Haemorrhage AHR—Adjusted Hazard Ratio, AOR—Adjusted Odds Ratio, ARR—Adjusted Risk Ratio, ART—Antiretroviral Therapy, BMI—Body Mass Index, BW—Birthweight, CFR—Case Fatality Rate, CS—Caesarean Section, EFV—Efavirenz, HAART—Highly Active Antiretroviral Therapy, IPI—Inter Pregnancy Interval, IPT—Intermittent Malaria Preventive Therapy, IPV—Intimate Partner Violence, LMP—Last Menstrual Period, LBW—Low Birthweight, MUAC—Mid-Upper Arm Circumference, NA—Not Applicable, NDD—Neurodevelopmental Delay, NR—Not Reported, NVP—Nevirapine, PIH—Pregnancy-induced Hypertension, PMTCT—Prevention of Mother To Child Transmission, PTB—Preterm Births, PROM—Premature Rapture of Membranes, RCT—Randomised Control Trial, SFH-Symphysis—Fundal Height, USS—Ultrasonography/Ultrasound, UTI—Urinary Tract Infection.

## Data Availability

The data supporting the conclusions presented in this article are available within this article and the Appendix A.

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
