# Peer review of "A Scoping Review of Preterm Births in Sub-Saharan Africa: Burden, Risk Factors and Outcomes"

_ijerph, 2022, doi:10.3390/ijerph191710537_

Round 1
Reviewer 1 Report
Please see the attachment.

Reviewer 2 Report
1. Using LMP is not a suitable method to estimate gestational age according the review. Is the real gestational age usually older or younger than the estimated age using this method? Please make it clear in the context.
2. Given the LMP method was unreliable, studies using this method might not able to calculate the amount of PTB population precisely, leading to misleading subsequent analysis and results. How about screening out the studies which used more reasonable GA method, checking this part of studies to see the risk factors involved? Please also specially discuss and compare the finding results from this small portion of studies.
3. As you can see, a lot of studies showed contrary results. Please give more details about the difference in the studies and possible causes when you describe or compare those studies.
4. In figure 2, please add space among the words of the title and the note “the highest number of publications”.
5. In abstract, the sentence “The estimated burden is potentially an under- estimation of the true burden given the widespread use of LMP, an unreliable and often inaccurate method for estimating gestational age” is not that easily understood. Can be written as “The true burden of PTB is underestimated due to the widespread use of LMP, an unreliable and often inaccurate method for estimating gestational age”.
6. There are some unnecessary blank spaces in the manuscript, such as page 2, line 71, 110, 255, 257, 259.
7. “hypertension in pregnancy” is repeated in the first paragraph of section 3.5.
8. Line 254, poor SSA should be “poor in SSA”.
9. Table 1 and 2 are detailed, but they are not that friendly read sorting by author initial of the publications. A summary table for risk factors and publications would be helpful (refer to this paper, DOI: 10.1136/bmjopen-2021-052576 )
Round 2
Reviewer 1 Report
The authors have satisfied all my comments and questions regarding their earlier manuscript and I believe this paper now merits publication. Thank you for the opportunity to review this work which highlights an important topic in the field of global child health.